# Co-Designing and Refining a Home-Based Exercise Programme for Adults Living with Overweight and Obesity: Insight from People with Lived Experience

Sofie Power * , Nikita Rowley, Michael Duncan and David Broom

Institute for Health and Wellbeing, Coventry University, Coventry CV1 5RW, UK; ad5173@coventry.ac.uk (D.B.)
* Correspondence: powers3@coventry.ac.uk

**Abstract:** Undertaking a home-based exercise programme should be a positive, health-enhancing lifestyle behaviour, particularly for adults living with overweight and obesity. However, exercise programmes are seldom designed in collaboration with people with lived experience, limiting adherence, efficacy, and effectiveness. Two focus groups (in-person $n = 6$ and virtual $n = 7$) were undertaken in the United Kingdom, to glean feedback and further refine a home-based exercise programme, developed from previously conducted semi-structured interviews with adults living with overweight and obesity. Both focus groups provided an opportunity for participants to discuss the proposed programme, highlighting strengths and areas for further improvement. Three key priorities were identified for consideration throughout the design process, specifically for adults living with overweight and obesity: (1) individualisation—a person-centred programme was non-negotiable; (2) motivation—integration of motivational features affected adherence and engagement; (3) more than just weight loss—consideration of other outcomes aside from solely numerical weight loss. These priorities provide direction for further refinement of the proposed home-based exercise programme, in an effort to ensure the final intervention is truly population-specific and needs-sensitive. Following completion, the programme will be assessed using a feasibility randomised controlled trial design.

**Keywords:** co-design; focus groups; home-based exercise; overweight and obesity

## 1. Introduction

Designing needs-sensitive, specific exercise programmes for population groups is fundamental to participants' continued engagement and long-term behaviour change [1]. Particularly for population groups that may further benefit from and prefer a tailored exercise programme [2], researchers should consistently listen to and consider participant needs and preferences throughout the design and development process. However, this has not always been the case up until more recently, when there has been an increase in the reporting of Patient and Public Involvement (PPI) within health and social care research [3]. Within person-centred research, integrating the voices of people with lived experience provides further insight than can be gleaned from the research team alone, an opportunity that would not be possible without undertaking PPI. In addition to meeting the requirements of research funders, this process further integrates members of the target population group, potentially increasing programme enrolment in comparison to outputs designed by researchers alone [4].

The involvement of members of the public through PPI is defined as research undertaken 'with' or 'by' service users, in contrast to and more commonly, undertaking research 'to' or 'for' population groups [5]. Co-production research approaches are more frequently being implemented within sports, exercise and health sciences, prompting the need for consideration of this approach and its use in the field so far [6]. Whilst this approach can be challenging to conduct, with time constraints [7] and the need to build trust with service users [8], the individualised, evidence-based output far outweighs the drawbacks.

However, to the author's knowledge, there is limited use of PPI when designing home-based exercise programmes. This is particularly important in the context of overweight and obesity, where the global prevalence of adults living with overweight and obesity is continually rising [9], and engaging and adhering to an exercise programme is fundamental to gaining associated health benefits.

Home-based exercise programmes may provide a facilitative solution. The home environment offers a flexible opportunity for adults living with overweight and obesity to exercise [10], with reduced feelings of weight stigma and shame that may be present if undertaking exercise at a leisure facility [11]. However, previously designed home-based exercise programmes have not always been population-specific. Often, signposting participants to resources available within the public domain, such as the Chief Medical Officers' Physical Activity Guidelines [12], when an individualised approach is often more suitable [13]. For adults living with overweight and obesity, the reduced consideration of individual circumstances may make engagement in exercise more challenging [14,15]. There should be a consideration of the influences of other lifestyle factors, such as socio-demographic status and varying home environments, subsequently creating sustained behaviour change using techniques in which the participant is individualised [16]. This highlights the PPI process as a tool to facilitate the consideration of external factors that researchers originally would not, and to provide an opportunity where participants are heard with the intention for action [17].

Home-based exercise has varying definitions within the physical activity and health literature; it is most commonly defined as exercise in a variety of different locations outside of a clinical setting [18]. However, there are many different exercise locations, delivery modes, and prescription types outside of a clinical setting to group all of them under the term 'home-based'. Therefore, following a search of the literature, discussion and expert opinion, we used the recently published definition of home-based exercise as 'exercise or physical activity undertaken inside or within the immediate vicinity of the home including the garden and/or driveway' [19] (p. 7).

Recently, we demonstrated through semi-structured interviews [20] that adults living with overweight and obesity would benefit from population-specific programmes. The present study aims to address the identified gap by providing a platform for feedback on the subsequently designed programme, identifying strengths and areas for further improvement. Secondly, we aimed to further develop and refine a needs-specific, population-sensitive programme in collaboration with people with lived experience.

## 2. Materials and Methods

Following ethics approval granted by Coventry University Ethics Committee (P133745), participants were recruited via email from previous research engagements and the National Institute for Health Research-People in Research website. Whilst ethics approval is not essential to conduct PPI, it was deemed good practice for full research transparency and required for publication. The purpose of this PPI was to glean insight from people with lived experience at a single time point, not to undertake a full qualitative study, as this had already been conducted to ascertain important elements of the home-based exercise programme from people living with overweight and obesity [20].

Thirteen participants were recruited and attended either an in-person ($n = 6$) or virtual focus group ($n = 7$) held between March and April 2022. All participants provided written informed consent prior to participation.

Both focus groups began with a short presentation by the lead researcher, detailing the results of the previously conducted interviews, and subsequently designed a home-based exercise programme, which included a virtual demonstration of the Hope digital self-management programme. This online platform aims to empower people to manage their health and well-being by providing knowledge, skills, and confidence to cope with different challenges they may be facing. Participants were provided with a demonstration of a self-management course centred around long COVID recovery; however, the platform can be

tailored to different health conditions, as reported within the literature [21]. Following a comfort break, the researchers facilitated a group discussion where participants discussed the proposed programme and online platform, highlighting positives and areas for further improvement and consideration. Prior to the discussion, participants were asked not to hesitate for fear of causing offence and reminded that there were no correct or incorrect answers, and that the researchers were interested in their views—positive, negative, or indifferent. The discussion guide created and used by the research team can be found in Supplementary Materials (S1).

The in-person event took place on the Coventry University campus in a pre-booked, clearly signposted room. The tables and chairs were arranged into a square to allow participants to see both the presentation screen and other group members. Refreshments were provided throughout the event, as well as easy access to toilets and associated facilities.

The virtual event took place using the Zoom software (version 5.9.1), with both researchers assigned joint control of the virtual room. All participants were allowed the choice of turning their cameras on or off, and provided their background was quiet, they could remain unmuted for the duration of the focus group. Following the comfort break in which participants could access their own refreshments if desired, all participants confirmed they had returned to the virtual room and that the event could continue.

Both focus group discussions were recorded via voice recorder (Sony ICD-PX370), and the virtual event was also recorded using the Zoom platform. The Zoom software chat record was downloaded, which accommodated neurodivergent participants during the discussion, who expressed a preference to contribute through written communication. The in-person event discussion was 1 h 22 min 39 s, and the virtual event discussion was 50 min 9 s.

Upon completion of the focus groups and following a preliminary data review, the facilitators determined there was no need for an additional focus group, having already gleaned rich insight and overlapping discussions between the two groups. It was also recognised that more participants within each group would have been unmanageable, potentially reducing opportunity for all participants to fully contribute as desired. Participants then received a £20 Amazon voucher for their time and engagement. Those that attended the in-person event also received reimbursement for their travel and associated costs. Participants were also requested to provide feedback on the event via a Google form.

Following each focus group, the lead researcher transcribed the recordings verbatim. Each transcript was re-read, and a summary table was created to highlight pertinent points raised by participants, and the subsequent changes to the home-based exercise programme. It was deemed unnecessary to analyse the focus group transcripts thematically. However, the research team is well-versed in this qualitative method from previous interviews [20]. For ease of interpretation and following discussion with the wider research team, we have synthesised the data into three key priorities to demonstrate the needs-sensitive design and development of this home-based exercise programme.

## 3. Results and Discussion

All 13 participants were adults aged $\geq 18$ years and volunteered for the study due to having lived experience of undertaking home-based exercise and/or living with or had lived with overweight or obesity previously. Due to the qualitative focus group research design, it was not deemed necessary to conduct a physiological sex analysis of the participants nor glean excess demographic data. The research team was interested in all participant feedback, regardless of additional participant demographics.

Three key priorities to consider throughout programme design and delivery were identified by the research team and presented by the participants of both focus groups: (1) individualisation, (2) motivation, and (3) more than just weight loss.

P*n* = participant number, V = participant attended the virtual event, and I = participant attended the in-person event.

### 3.1. Priority One—Individualisation

The overarching priority presented during both the in-person and virtual event was the importance of individualisation, and there was a strong, almost non-negotiable desire that the programme should facilitate this:

*"You're talking about people living their lives. That's the whole point isn't it, it's person centred, good things and bad things happen so you've got to have that flexibility is the key to it really."* (P5,I)

This individualisation referred to both the malleability of the delivery platform and the content of the programme itself. For example, designing a programme that can facilitate the adaptation of exercises to the surrounding environment, or how the participant feels on that day, recognising that this would be different for each person:

*"The right fit for you might not be someone like, I've said it as well, it's not necessarily someone that's the same as you."* (P1,I)

Although, both groups of participants consistently acknowledged that the feasibility of programme individualisation to their desired standard would be challenging:

*"I know what it's like developing these things, that the the the list of um your wants and what can be delivered straight away is two different things."* (P3,I)

This recognition was more frequent within the in-person discussion, where all participants, but one, had undertaken the semi-structured interviews that tailored the design of the proposed programme. Throughout the discussion, the participants recognised the complexity of their requests within the constraints of a time-restricted research project, a consideration previously recognised by researchers within PPI associated literature [7]. Despite this understanding from participants that their suggestions may not be feasible or would lead to further programme considerations for the research team, the opportunity for constructive feedback is a process that should be welcomed and encouraged within any PPI [22].

Without this individualisation priority, participants expressed a lack of desire to undertake a home-based exercise programme. Without tailoring, they felt the programme would not be effective for them, and there was nothing to separate it from pre-existing resources. This demonstrates participants' recognition of the gap within currently available home-based exercise programmes, specifically for adults living with overweight and obesity, and subsequently identifying individualisation as a priority that is missing. The process of leading a healthier lifestyle, whether this was associated with weight loss or not, was consistently recognised by participants as an individual journey, with each person having different goals and motivations for change:

*"It's very specific to you rather than having a general um um general outcomes because exercise and health is very individual and very you know uh idiosyncratic, and what you like and what you don't like and what's important to you."* (P5,I)

Participant desire for individualisation within the exercise programme design is repeated throughout the paper and intertwines with the other two identified priorities. Particularly for adults living with overweight and obesity, many have already attempted multiple exercise programmes as a means to lead a healthier lifestyle, but without long-term success, arguably due to a lack of individualisation [23]. Therefore, it would be important to consider the reasons for this absence of long-term behaviour change. If people with lived experience are presenting the need for individualisation, this may be a consideration that has not already been successfully implemented, justifying the need for a programme to have the capacity to do so. If a greater focus on individualisation can lead to increased participant engagement and adherence, this can then contribute to longer-term behaviour change in participants, through consistently undertaking exercise that is suitable for their needs. Regardless of the logistical implications for researchers and practitioners, if individualisation facilitates this, it should be prioritised throughout programme design and delivery.

*3.2. Priority Two—Motivation*

Consideration and integration of programme features to motivate participants to engage and adhere to the home-based exercise programme were important. However, supplementary to the individualisation priority, participants recognised that features and techniques that might be motivating for one might not be motivating for another:

*"So the motivations in this room will be different."* (P1,I)

Despite this, it was important for the programme and the delivery platform to include elements that could offer motivational techniques for the majority of participants. Without this, participants struggled to visualise their long-term engagement in the programme. For some, this included motivation to initiate their participation with a home-based exercise programme, not just motivation for continued engagement:

*"It could be the first step someone is taking in trying to lose weight or anything of the kind and they need to have motivation to carry on, that structure and that plan in place."* (P4,V)

Participants discussed and debated how motivational strategies could be integrated into an online platform, particularly because the more commonly used techniques required tangible materials that could not be virtually replicated. One participant spoke about using object prompts in their daily life as a reminder of why they were trying to increase their physical activity:

*"So sometimes when I have like oh I've got a birthday coming up, you know, and I would hang out one of my clothes that I'd want to wear and that would be an inspiration for me."* (P2,V)

Whilst an online, virtually delivered programme would present motivational challenges for implementing techniques such as physical prompts and cues, this may not mean that a virtual programme would be unsuitable for all participants just because the motivational element is not tangible [24]. This prompts consideration for both the type of motivational techniques that are implemented, and the way in which they are integrated when designing and developing home-based exercise programmes. It may also lead to research team discussion and the provision of multiple different techniques for participants to choose from, in order to build a programme that is needs-sensitive.

Participants expressed desires for a variety of motivational techniques, whilst remaining aware of individual preferences, but still recognising that the provision of any motivational techniques, such as incentives, for programme engagement would be more helpful than unhelpful [25]. For example, some participants wanted pre-session reminders throughout the duration of the programme:

*"You decide you want to do the class at seven o'clock on a Tuesday and an email comes through at six o'clock saying... it might not necessarily work but it might help some people you know saying your your class is coming up, so it's your own self-imposed dead self-imposed goal."*(P3,V)

Whereas others wanted to receive post-session congratulations, reinforcing justification for their participation in the programme and encouraging continuation:

*"After a class, if I've completed it, it depends again not everyone wants emails coming in all the time but well some something for me it might be nice to get an email saying well done it's great that you attended."* (P3,V)

At the in-person event in particular, participants spoke about a buddying system where programme participants could be paired up, derived from demographic characteristics, motivations, and goals, developing accountability and motivation through a shared journey and collective identity:

*"Somebody else on the programme who was specifically your buddy who you could support. And by you know that if you have a naff week then you know you can say to*

*them yeah I've had a naff week or they can say to you I've had a naff week and you can say oh that's ok because I did last week but look I'm back on it.* (P2,I)

This also provided a space where participants could engage in the programme, remaining aware of their individual progress but without placing emphasis solely on themselves:

*"I'm not then just focusing on my on myself. And then if I don't necessarily do it one week, it's not a problem because someone else might pick up the mantle in terms of the contributing."* (P1,I)

This would also provide a social element that was greatly desired as a result of the semi-structured interviews [20], provided the buddying pairs are made appropriately. Whilst proven to be an effective method to increase physical activity [26], individuals are not all motivated in the same way, and incompatible partnering may hinder motivation and, consequently, programme engagement. Therefore, integrating a range of tailorable, varied techniques within the programme could be considered a favourable design practice for programme motivation. Regardless of the other population group characteristics, what motivates one individual to exercise is not guaranteed to motivate another [27]. Hence, further highlighting the influence of, and need to, consider participant individuality within programme design to best accommodate the complexity of lived experience needs.

### 3.3. Priority Three—More Than Just Weight Loss

Particularly at the in-person event, although present at both, participants consistently spoke of other programme characteristics and benefits to consider during the design and delivery process, aside from solely numerical weight loss. For most, increasing their exercise and decreasing sedentary behaviour is a long-term lifestyle change, and participants expressed preferences to reduce the focus on weight loss because that was not their primary reason for participation:

*"My overwhelming aim is to get healthier, because then other things will follow... If I had to pick one word why I would be willing to give this a go, it's for the benefit of my health."* (P4,I)

This preference for a holistic, lifestyle-focused approach is also demonstrated quantitatively within the literature, where physical activity has been proven to be beneficial for participants, regardless of numerical weight loss [28]. This parallel between participant desires and supporting research evidence may provide reassurance to participants that their desires are also beneficial for their health. It also provides a rationale for continuing the integration of PPI within programme design and development, allowing for the relationship between research and lived experience to continue to grow.

Considering the wider impact of physical activity, further than just weight loss, was also reflected in comments regarding the exercise terminology used throughout the programme proposal. Specifically for some participants, just engaging and adhering to the programme was an achievement, regardless of the physiological health outcomes, and they wished for the programme to recognise that:

*"Could you um not use regression... because it sounds like a definite failure word... if you're having a bad day like... say with whatever condition, you're already having a bad day so to then thinking oh I've regressed, can't do as opposed to, these are the alternatives."* (P2,I)

The development and delivery of a programme that focuses on more than just weight loss was predominantly presented by participants at the in-person event. This may be a result of their investment in the programme from the semi-structured interview phase, and therefore, the focus group discussion thoroughly explored finer details, such as programme language and purpose. However, at the virtual event, participants were newer to the programme and still familiarising themselves with the aims and purpose; therefore, their focus group discussion was more objective, exploring the programme and delivery platform from an outside perspective. These two groups allowed for input and refinement from different

depths of programme investment with people with lived experience, a consideration that should be given thought to when conducting PPI [29]. Whilst this may not always be feasible or applicable, the inclusion of people with different lived experiences and varying prior knowledge will contribute towards a more informed programme.

Both groups of participants mentioned the importance of considering other programme characteristics in addition to short-term participant engagement and enjoyment. They presented the idea that including a more holistic approach to programme outcomes would positively influence their programme uptake and engagement. For example, to design and develop an informed programme, participants considered emotional well-being to be of equal importance to the physiological programme measures and wished for this to be consistently integrated:

*"Be aware of it (emotional wellbeing) as it goes along... at the end of the 12 weeks is probably when they're more likely to say something than you know on day one."* (P6,I)

This was reinforced by the importance of facilitating the monitoring of emotional well-being alongside monitoring physiological measures:

*"For me when I do these exercises, I feel more radiant... how do I measure that?"* (P2,V)

Participants recognised that the provision of this support might be beyond the scope of the programme, and therefore sufficient signposting should be available to those that may need additional help:

*"I think you've raised a really good point. Signposting to professional services is is a must."* (P3,I)

Considering a wider range of lifestyle outcome measures within the programme design, as well as signposting to external resources, benefits the participant and the programme designers in multiple ways. Firstly, through the provision of further relevant resources that participants may find helpful. Secondly, it demonstrates that the programme developers have listened to the feedback from the focus groups, having recognised and considered the link between emotional and physical health. The positive impact on participants of acting upon their suggestions from PPI events has been previously reported within the literature [30]. These may include participants feeling like being part of a team and provided the opportunity to contribute towards problem solving, both of which would be important aims for researchers and practitioners to foster when undertaking effective PPI.

In addition to the three priorities identified above, participants also spoke about other considerations, such as the relatability of the exercise instructor and how that influenced their ability to engage, sharing participants' success, and how an online programme could do this without eliciting negative feelings for others. Whilst these are just a few examples highlighted by participants, Table 1 displays additional participant suggestions alongside corresponding programme changes.

Of the participants that completed the post-focus group feedback forms (*n* = 6), 100% stated that they had the opportunity to contribute as desired and were given appropriate time to respond to questions.

**Table 1.** Participant programme preferences with supporting quotes and corresponding programme alterations.

| Participant Discussion Point | Supporting Quote | Corresponding Programme Response |
|---|---|---|
| Programme facilitators/deliverers should be people with lived experience. | *"You can't have somebody then saying I understand, when they're slim, they're fit and they're healthy and they go to the gym ten times a week . . . so actually to have somebody who it who is the same, I think is a real win."* (P2,I) | The programme materials (videos and images used) will be representative of adults living with overweight and obesity. |
| Exercises should have adaptations/alternatives, not 'regressions', to accommodate individual circumstances and abilities, avoiding demotivation from finding some exercises 'too hard'. | *"You're already having a bad day so to then thinking oh I've regressed, can't do, as opposed to these are the alternatives."* (P2,I) | Terminology used throughout the programme will be carefully considered, ensuring it is sensitive to the population group. |
| Desire to receive optional prompts/reminders throughout the week, concern about forgetting and reaching the end of the week with sessions to catch up on. | *"Maybe you could set or at some point the system could set some form of notification."* (P3,V) | Dependent on the platform, sending prompts/reminders may be feasible, and we would aim to do so. |
| Desire to receive optional rewards/congratulations upon session and goal completion to increase motivation and adherence. | *"Something for me, it might be nice to get an email saying well done it's great that you attended."* (P3,V) | Dependent on the platform, we will aim to send rewards/congratulations upon session completion. |
| Creation of an ethos where participants do not have to start over if their engagement does not go 'perfectly'. minimising one missed session snowballing into complete 'programme failure'. | *"If you're a little bit sensitive or you know you've had a few failures in previous lives then you know you would get to that stage and think, I've blown it again, and not actually go back."* (P3,V) | To consider and integrate into the programme materials, emphasising participants tailoring the programme to themselves. |
| Flexibility to alter the online platform features to suit them best and maintain desired privacy. | *"So that it's a very personably built rather than a generic platform."* (P2,I) | Whilst it may not be feasible to alter the features per participant, we can make help participants choose the features that work best for them. |
| Creation of a buddy system to share knowledge and experience and create accountability for increased motivation. Bridging the gap between participants' high motivation for social exercise and low motivation for exercise that focuses on individual improvement. | *"You're doing it to help yourself but your main focus is somebody else."* (P6,I) | A potential feature that was highlighted prior and reinforced by the PPI. Pairings and logistics will need to be carefully considered. |
| Peer support network, in addition to a buddy system, as a space where participants can share their progress, provide tips, and congratulate others if they wish. Potential for this to be in the form of a virtual call or an online platform. | *"It's important after three weeks everyone gets together, tells you how they're feeling, brainstorming, negative, positive feedback, any tips and hints people have and can give each other to encourage each other to carry on with it."* (P4,V) | A suggestion that could be implemented easily, facilitated by the rise in use of virtual meetings and the forum facility already built on the online platform. |
| Focus on the overall health benefits rather than just weight loss, with appropriate signposting to additional resources available to all. Being able to measure these changes alongside the physiological measures. | *"My overwhelming aim is to get healthier, because then other things will follow."* (P4,I) | Reflects the programme's aim for participants. Considerations to be made regarding measuring emotional well-being. |

**Table 1.** *Cont.*

| Participant Discussion Point | Supporting Quote | Corresponding Programme Response |
|---|---|---|
| Longer duration than 12 weeks. Expressed a desire to access the programme after 12 weeks to go back, continuing to build a habit. | *"Twelve weeks is lovely except that it's not a life changer."* (P2,I) | Continued access to programme materials and resources could be arranged for participants after the trial has finished. |
| Current programmes focus on individuals that heavily promote their own success. Concern was presented regarding comparison of successes despite the variation of individual goals and how different goals could be catered for. | *"People are gonna take a lot more notice of a ten stone weight loss than they are to the person that's lost their first pound . . . those who shout the loudest get the most attention."* (P1,I) | Consideration required regarding the online platform privacy. A virtual reward system to focus on goal achievement rather than goal content or size. |
| Concerns of safety within the home environment and precautions needed to restructure the environment to accommodate this. | *"At that set up point, if if you're gonna be doing weights or whatever you're doing you know you need that sort of space"* (P2,I) | Provision of information and reminders regarding safety considerations and minimising risk in their home environment. |
| Questions regarding the provision of equipment: reliability, quality, and cost. Emphasis placed on capitalising on materials in the home environment because it is a home-based programme. | *"If it's home-based try and make the best use of what you've got in your house."* (P4,I) | Chosen equipment will be minimal and provided by the research team. Alternatives found in the home will also be suggested within the programme material. |
| Education on exercise types and benefits. Participants wanted to know what type of exercise 'counted' and the benefit each exercise has. | *"When I do a pilates class once a week and the instructor always tells us which bit of our body it's helping with, and that actually is quite an interesting useful thing."* (P2,I) | Inclusion of educational components and justifications within the exercise videos and additional programme resources. |
| Questions regarding accessibility include digital literacy, language barriers, neurodiversity, and socio-economic class. | *"Would there be financial help to support internet costs (during the trial)? Would you have translation into other languages on the website?"* (P5,V) | Whilst it was recognised that these considerations are very important, they are not feasible to facilitate within the scope of the current project due to time and resource constraints. However, these issues will be considered should the project be run on a larger scale. |
| Requirement for at least some in-person contact throughout the duration of the programme to feel supported and reduce potential dropout. | *"The fact that an actual person knows what their plans are, it might really motivate them."* (P3,V) | Baseline, midpoint, and endpoint data collection sessions will be in person. Additional to the researchers collecting data, it will allow for in-person interaction to increase feelings of support and connectedness. |
| Further involvement in pilot testing—both for the programme itself and the delivery platform. | *"Well that would be brilliant, yes I would like that."* (P3,V) | Volunteers at the PPI events will be offered to join the pilot. |

P*n* = participant number, V = participant attended virtual event, I = participant attended in-person event. PPI = Patient and Public Involvement.

### 3.4. Strengths and Limitations

Using PPI to highlight considerations regarding the design and delivery of home-based exercise, specifically for people with overweight and obesity, has to the author's knowledge, not previously been utilised. Consequently, the use of this approach should be considered a strength of the work, and a consideration for other researchers when developing exercise programmes.

It should be noted that participants verbally confirmed their lived experience of the topic; however, what was deemed to constitute lived experience was not explicitly defined or screened for in participants prior to the discussion. Although, what may qualify as enough lived experience remains unclear [31]. Despite this, the authenticity of participants' stories was consistently demonstrated by the participants, and can be attested to by the two members of the research team conducting the focus groups.

Conducting data collection in two settings was a key strength. It accommodated participant engagement preferences and limited withdrawal due to reasons such as travel time and expense. Whilst it could be considered that the between-participant discussion may have flowed more during the in-person event due to the reliance on technology for the virtual event, the opportunity for between participant discussion was still provided for both groups. The virtual event also facilitated participants that preferred to contribute to the discussion in a written manner, and for further contributions through the chat function, without interrupting the verbal discussion. Additionally, whilst restrictions relating to COVID-19 had been lifted at the time of data collection, we had to be mindful that there is still some hesitancy regarding face-to-face meetings. So offering multiple settings for data collection facilitated a larger participant number, which is important within a PPI process.

A potential limitation is that a focus group setting may negatively influence the willingness of participants to express their views on a sensitive topic. However, a specific setting was required in order to create between-participant discussions that would not have been present in an interview. Participants were also given the opportunity to anonymously provide feedback regarding presenting their views and opinions during the event in the feedback form, of which no negative feedback regarding the discussion environment was received.

All but one of the participants at the in-person event contributed towards the design of the home-based programme during the semi-structured interview phase. This may have influenced their discussion responses for fear of offence to the research team or a greater understanding of the limitations of the programme design process. In an attempt to manage this, participants were reminded that the research team was interested in all of their views, positive and negative, before the discussion commencement. Additionally, none of the virtual participants had undertaken the semi-structured interviews, and therefore a balance was achieved between participants with and without programme design and delivery preconceptions.

During the event and in the feedback form, many participants expressed a desire to contribute towards pilot testing of the home-based exercise programme. This suggests the research team had built an environment in which participants felt valued and able to trust [8], leading to a willingness to invest further time in the future development of the programme.

### 3.5. Future Development and Practical Application

Specifically for this project, the refined programme will be piloted to glean further feedback, both regarding the application of the feedback received from the PPI events, as well as the usability and suitability of the programme going forward. The subsequently designed programme will then be tested as part of a feasibility RCT.

Whilst we recognise that PPI can be challenging and time-consuming to conduct, particularly in time-constrained research, the benefits to both the participants and the researchers far outweigh the drawbacks. Therefore, there should be a continued increase

in the involvement of people with lived experience, through PPI processes, during the development of exercise programmes for specific population groups.

## 4. Conclusions

This research aimed to collect and collate feedback regarding the development of a home-based exercise programme specifically for adults living with overweight and obesity. Three key priorities were identified as programme design considerations: (1) individualisation—a person-centred programme was non-negotiable; (2) motivation—integration of motivational features affected adherence and engagement; (3) more than just weight loss—consideration of other outcomes aside from solely numerical weight loss. This further tailoring of the designed home-based programme will result in a needs- and population-specific programme developed in collaboration with people with lived experience. Something that, to the author's knowledge, has not been undertaken previously with this specific population group and exercise location, and will go on to be tested as part of a feasibility RCT. Generally, it will also provide direction for more refined and informed population-specific exercise programmes, by creating opportunities for people with lived experience to have their voices heard and acted upon.

**Supplementary Materials:** The following supporting information can be downloaded at https://www.mdpi.com/article/10.3390/obesities3020011/s1, S1: Discussion Guide.

**Author Contributions:** Conceptualisation, S.P., M.D. and D.B.; methodology, S.P. and D.B.; investigation, S.P. and D.B.; resources, S.P.; data curation, S.P.; writing—original draft preparation, S.P.; writing—review and editing, S.P., N.R., M.D. and D.B.; visualisation, S.P., N.R., M.D. and D.B.; supervision, N.R., M.D. and D.B.; project administration, S.P.; funding acquisition, M.D. and D.B. All authors have read and agreed to the published version of the manuscript.

**Funding:** This research was funded by Research England provided to Coventry University to support participatory research, grant number N/A.

**Institutional Review Board Statement:** The study was conducted in accordance with the Declaration of Helsinki and approved by the Institutional Ethics Committee of Coventry University (protocol code P133745 and date of approval 22 February 2022).

**Informed Consent Statement:** Informed consent was obtained from all subjects involved in the study.

**Data Availability Statement:** The anonymised data presented in this study are available upon reasonable request to the corresponding author. The full data are not publicly available to maintain participant anonymity.

**Acknowledgments:** The authors would like to thank all the participants that took the time to participate and engage in the focus group events and the technician support throughout the programme development.

**Conflicts of Interest:** The authors declare no conflict of interest. The funders had no role in the design of the study, in the collection, analyses, or interpretation of data, in the writing of the manuscript, or in the decision to publish the results.

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
