# Peer review of "Co-Designing and Refining a Home-Based Exercise Programme for Adults Living with Overweight and Obesity: Insight from People with Lived Experience"

_2673-4168, doi:10.3390/obesities3020011_

Round 1
Reviewer 1 Report
In the abstract, it should be added that the research was carried out in the United Kingdom.
In the part of material and methods, the characteristic of focus groups are insufficient. Where they overweight or obese? If so, what criteria were used to define overweight or obesity? It was only indicated, that they was people aged 18 or more. What was the age range, average age?
If the authors collected sociodemographic data (e.g. education, place of residence), they could also be provided.
The conclusions are very general. They should be more precisely defined.
Reviewer 2 Report
· Your article seems to be fine and original.
· How was the sample size calculated?
· What was the criteria to choose people with lived experience?
· Page 5 line 194-195: Please clarify the point you are attempting to make. "long- term behaviour change, this should be prioritised throughout programme design and delivery"
· Could you explain more about how to design exercise programs, and how to conduct physiological sex analyses of the participant when collecting and collating feedback regarding the development of whom based exercise program.
· The idea behind this work is good but discussion needs precision and clarity. Could you elaborate more on this by specifically talking about identified gaps, develop and refine needs specific in collaboration with people with lived experience.
